# Ginkgolic acid attenuates *echinococcus granulosus* infection-induced hepatic fibrosis by inhibiting Smad4 SUMOylation

Qiuyue Chen[1,3,4☯], Dan Dong[1,3,4☯], Xueting Yu[1,5], Xinyu Jiang[1,2,3,4], Huijiao Jiang[1,3,4], Jun Hou[1,3,4], Lianghai Wang[1,3,4], Junying Xu[1,3,4], Xiangwei Wu[1,2,3,4*], Xueling Chen[1,3,4*]

**1** Shihezi University School of Medicine, Shihezi, China, **2** The First Affiliated Hospital of Shihezi University, Shihezi, China, **3** NHC Key Laboratory of Prevention and Treatment of Central Asia High Incidence Diseases, Shihezi, China, **4** The Clinical Research Center for Infectious Diseases of Xinjiang Production and Construction Corps, Shihezi, China, **5** Huludao Center for Disease Control and Prevention (Municipal Institute of Health Supervision), Liaoning Province, China

☯ These authors contributed equally to this work.

\* wxwshz@126.com (XW); chenxueling@shzu.edu.cn (XC)

## Abstract

### Background

The SUMOylation modification is closely linked to the progression of fibrotic diseases, yet its role in hepatic fibrosis associated with cystic echinococcosis (CE) remains unclear. This study aimed to investigate the function of SUMOylation in CE-related hepatic fibrosis and evaluate the anti-fibrotic effects and mechanisms of ginkgolic acid (GA) via regulation of the SUMOylation pathway.

### Methodology

Peri-lesional (PL) and adjacent normal (AN) liver tissues from CE patients were collected to examine histopathology and SUMO pathway proteins. A CE-infected mouse model was established and treated with GA to assess cyst burden, serum TGF-β1 levels, hepatic fibrosis markers, and SUMO-related proteins. *In vitro*, macrophages and hepatic stellate cells (HSCs, LX-2 line) were stimulated with *Echinococcus granulosus* cyst fluid (EgCF) or TGF-β1 to evaluate GA's effects on macrophage polarization (CD206/CD86), HSC activation (α-SMA/PCNA), Smad4 SUMOylation, and nuclear translocation. Macrophage-HSC crosstalk was investigated via conditioned medium co-culture assays.

### Result

Fibrosis was exacerbated in peri-lesional liver tissues of CE patients, accompanied by SUMO pathway activation. GA significantly alleviated hepatic fibrosis in CE mice and reversed SUMO pathway dysregulation. Mechanistically, GA inhibited

**Data availability statement:** All relevant data are within the manuscript and its Supporting Information files.

**Funding:** This study was supported by the National Key R&D Program of China (2024YFC2309700 to XC); General Program of National Natural Science Foundation of China (82573336 to XC); Corps Guidance Science and Technology Plan Project (2024ZD018 to DD, 2023ZD034 to XW); Tianshan Young Talent Scientific and Technological Innovation Team (2023TSYCTD0020 to XW); Shihezi University Scientific Research Project (ZZZC2023026 to HJ). The funders had no role in study design, data collection and analysis, decision to publish, or preparation of the manuscript.

**Competing interests:** The authors have declared that no competing interests exist.

EgCF-induced pro-fibrotic M2 macrophage polarization and blocked Smad4 SUMOylation and nuclear translocation by modulating SUMOylation. Furthermore, GA directly suppressed HSC activation and bidirectionally disrupted the pro-fibrotic crosstalk between macrophages and HSCs under EgCF stimulation, ultimately alleviating fibrosis.

## Conclusion

This study reveals the critical role of SUMOylation modification in CE-associated hepatic fibrosis and elucidates a novel anti-fibrotic mechanism whereby GA targets the SUMOylation-Smad4 axis to regulate the immune microenvironment.

### Author summary

This study demonstrates aberrant activation of the SUMOylation pathway during hepatic fibrosis progression in cystic echinococcosis (CE), characterized by up-regulated SUMO1/Ubc9 expression and downregulated SENP1 in peri-cystic liver tissue. Ginkgolic acid (GA) intervention significantly attenuated CE-associated liver fibrosis in mice, evidenced by reduced cyst volume, decreased TGF-β1 levels, and suppressed expression of fibrotic markers (α-SMA/COL1A1). GA concurrently reversed dysregulated SUMO pathway protein expression. Mechanistically, GA upregulates the deSUMOylating enzyme SENP1, thereby inhibiting Echinococcus granulosus cyst fluid (EgCF)-induced SUMOylation and nuclear translocation of Smad4. This blockade impedes macrophage polarization toward the pro-fibrotic M2 phenotype (CD206↓) and suppresses hepatic stellate cell (HSC) activation (α-SMA/PCNA↓). Furthermore, GA disrupts the pro-fibrotic bidirectional crosstalk between HSCs and macrophages (Fig 1). Collectively, these findings indicate that GA ameliorates CE-induced hepatic fibrosis by targeting the SUMO-Smad4 axis to modulate the immune microenvironment, providing a novel therapeutic strategy.

## Introduction

Cystic echinococcosis (CE), also known as hydatid disease, is a zoonotic parasitic disease caused by the larval stage of *Echinococcus granulosus sensu lato* (s.l.). It poses a serious threat to public health, especially in developing countries with a well-developed livestock industry, such as northwestern China [1,2]. Thus, CE is considered as a neglected tropical diseases by the World Health Organization [3]. The liver is the primary target organ, where the developing hydatid cyst induces pathological hepatic fibrosis through persistent mechanical compression and chronic immune-mediated inflammation, driving cycles of tissue damage and repair [4]. This resultant hepatic fibrosis represents a critical pathological juncture in the progression of various chronic liver diseases toward cirrhosis and eventual liver failure [5].

**Fig 1. Schematic representation of the proposed mechanism by which GA alleviates hepatic fibrosis in cystic echinococcosis through inhibition of Smad4 SUMOylation.** (Created by Figdraw.com with Copyright Code: RAIUA88388).

Therefore, elucidating the molecular mechanisms underlying *E. granulosus*-induced liver fibrosis is crucial for developing novel anti-fibrotic therapeutic strategies.

The central event in hepatic fibrosis is the activation and proliferation of hepatic stellate cells (HSCs) [6,7]. Under stimulation from various injurious factors, quiescent HSCs transform into activated myofibroblasts, which extensively express α-smooth muscle actin (α-SMA) and secrete extracellular matrix components, primarily type I collagen (COL1A1), leading to the formation of fibrotic scars [8–10]. In the microenvironment of *Echinococcus granulosus* infection, the antigen-rich hydatid cyst fluid as a key driver of persistent inflammatory and fibrotic responses [11–12]. Among immune cells, macrophages, as central players in innate immunity, can polarize into pro-inflammatory (M1-type, high CD86 expression) or pro-fibrotic (M2-type, high CD206 expression) phenotypes in response to different stimuli. By secreting cytokines such as TGF-β1 and IL-13, they play a pivotal role in regulating HSC activation [13–14]. However, the specific molecular mechanisms by which *Echinococcus granulosus* regulates macrophage polarization and thereby drives HSC activation remain to be fully elucidated.

In recent years, the role of post-translational modifications in cellular signal transduction has garnered increasing attention. The small ubiquitin-like modifier (SUMO)ylation is a dynamic and reversible modification process, similar to ubiquitination. It is catalyzed by the SUMO-activating enzyme E1, the conjugating enzyme E2 (UBC9), and ligases E3, and can be de-SUMOylated by SUMO-specific proteases (SENPs) [15–16]. SUMOylation alters the stability, subcellular localization, and transcriptional activity of target proteins, and is extensively involved in the pathogenesis of inflammation, stress responses, and fibrotic diseases [16–19]. The transforming growth factor-β (TGF-β)/Smad signaling pathway is the most classical pro-fibrotic pathway in hepatic fibrosis. Smad4, as a common mediator molecule, has been shown to undergo its SUMOylation modification has been proven to affect its nuclear translocation and transcriptional activity, and may be a key

node connecting inflammatory signals and fibrotic responses [20–21]. However, the role of SUMOylation in *Echinococcus granulosus* infection-induced hepatic fibrosis, particularly in the regulation of macrophage function, remains unclear.

Ginkgo acid (GA) is a kind of small molecular phenolic organic compound, and many of its functions are closely related to SUMOylation [22–23]. In recent years, it has been found that it has many pharmacological activities such as anti-tumor, anti-virus, anti-inflammatory and anti-fibrotic [24–25]. However, the impact of GA on hepatic fibrosis induced by CE remains unclear, and its underlying mechanisms of action warrant further investigation.

Based on the above background, this study proposes the scientific hypothesis that in the microenvironment of *Echinococcus granulosus* infection, hydatid cyst fluid may promote the Smad4 SUMOylation, thereby exacerbating the pro-fibrotic polarization of macrophages and driving HSCs activation; whereas GA may exert its anti-fibrotic effects by intervening in this process. We first observed in patient liver tissues that the expression of key SUMOylation modification proteins (SUMO1, UBC9, SENP1) was dysregulated around the lesions and correlated with the levels of fibrosis markers (α-SMA, COL1A1). In the infected mouse model, GA intervention significantly modulated these SUMO pathway molecules and effectively suppressed the progression of liver fibrosis. Further *in vitro* experiments demonstrated that GA, by regulating the SENP1/SUMO1 balance, inhibited the Smad4 SUMOylation and its nuclear translocation by the cyst fluid in macrophages, thereby reversing their pro-fibrotic polarization and blocking their fibrotic promotion of HSCs. This study not only provides novel theoretical insights into the pathogenesis of hepatic fibrosis in CE, but also lays a solid experimental foundation for developing GA as an adjunctive therapeutic agent for this disease.

## Methods

### Ethics statement

The study strictly followed the relevant ethical guidelines. All clinical samples were obtained after written and signed informed consent, which was approved by the Ethics Committee of Science and Technology of the First Affiliated Hospital of Shihezi University (approval number: KJ2025-003-01). The experimental procedures involving animals followed the principles of animal welfare and were reviewed and approved by the Bioethics Committee of Shihezi University (approval number: A2024-427).

### Chemicals and reagents

GA (MCE, China), β-mercaptoethanol (Gibco, USA), phorbol 12-myristate 13-acetate (PMA) (MultiSciences, China), Lipopolysaccharides (LPS) (Solarbio, China), recombinant human TGF-β1 protein (Active) (Abcam, USA), INTERFERin siRNA transfection reagent (NoninBio, China). All reagents were reconstituted and used according to the manufacturers' instructions.

### Preparation of *Echinococcus granulosus* protoscoleces and cyst fluid

Protoscoleces (PSC) and cyst fluid of *Echinococcus granulosus* were obtained from hepatic hydatid cysts of naturally infected sheep collected from a slaughterhouse in Shihezi, Xinjiang, China. Briefly, the liver surface was disinfected with 75% ethanol. A sterile 50 mL syringe was used to puncture the cyst and aspirate the cyst fluid containing PSC into a sterile centrifuge tube. The mixture was allowed to stand for 15 min to allow the PSC to sediment. The clear supernatant was centrifuged at 1000 g for 10 min, filtered through a 0.22 μm membrane, and then aliquoted and stored at –80°C for later use after protein concentration determination. The sedimented PSC were washed three times with sterile PBS. Viability of the PSC was assessed using the eosin exclusion assay, in which live PSC remain unstained while dead or damaged PSC stain red. Only preparations with viability exceeding 90% were used for subsequent experiments.

## Animal experiments

Female C57BL/6 mice were purchased from SpePharm (Beijing) Biotechnology Co., Ltd. (Laboratory Animal Production License: SCK (Jing) 2019–0010). After one week of acclimatization under controlled conditions (22°C - 24°C, 40%-60% humidity, 12/ 12-hour light/dark cycle), the experiment commenced. Thirty-two mice were randomly divided into four groups (n = 8): vehicle control group: Intraperitoneal (i.p.) injection of an equivalent volume of saline; GA safety group: i.p. injection of GA (25 mg/kg/day); CE group: *Echinococcus granulosus* (hydatid disease) infection mode; CE + GA Group: *E. granulosus* infection combined with i.p. injection of GA (25 mg/kg/day) [22]. The mice in CE group and CE + GA group were anesthetized, and inoculated with PSCs (3000 per mouse) by subcapsular injection, as described previously [26]. Eight weeks after infection, the GA and CE + GA groups were intraperitoneally injected with GA (25 mg/kg/d), and the solvent control group was injected with the same volume of normal saline. After 4 weeks of continuous administration, all mice were euthanized, and serum and target organ tissues were collected for subsequent experiments.

## Histopathological and immunohistochemical Staining

The liver tissue samples were fixed with 4% paraformaldehyde, embedded in paraffin, and cut into 4 μm thick sections. After routine dewaxing and hydration, the following staining methods were performed: (1) Hematoxylin-eosin (HE) staining was used to evaluate the histological changes of the tissue; (2) Masson trichrome staining and Sirius Red staining were used to observe the degree of fibrosis; (3) For immunohistochemical staining, sections were processed through antigen retrieval, endogenous peroxidase blocking, and serum blocking, the primary antibody (S1 Table) was incubated at 4°C overnight, and then the IHC two-step kit (Zhongshan Bridge, China) was used for color development and hematoxylin re-staining. Among them, the positive areas of Masson staining, Sirius Red staining and immunohistochemical sections were quantitatively analyzed using Image J software to compare the fibrosis levels and the expression differences of the target protein among different groups.

## Tissue Immunofluorescence Assay

After sequential steps of deparaffinization, rehydration, antigen retrieval, and endogenous peroxidase inactivation, the sections were incubated with primary antibodies (S2 Table) overnight at 4°C. Subsequently, corresponding fluorescent secondary antibodies were applied and incubated at room temperature in the dark. Nuclei were stained with DAPI. Finally, fluorescence microscopy (NIKON ECLIPSE C1, Japan) was used to observe the staining results and collect images.

## Enzyme-linked immunosorbent assay (ELISA)

Mouse serum TGF-β1 levels were quantified using a mouse-specific TGF-β1 ELISA kit (JL12223-48T, JONLNBIO, China), while IL-13 and TGF-β1 levels in cell supernatants were measured using a Human IL-13 ELISA Kit (EK113, MULTISCIENCES, China) and a Human TGF-β1 ELISA Kit (EK981, MULTISCIENCES, China), respectively; all assays were performed in strict accordance with the manufacturers' instructions.

## Cell experiments

Human hepatic stellate cells (LX-2), human monocytic cells (THP-1), and mouse monocyte-macrophage cells (RAW 264.7) were obtained from the Cell Bank of the Chinese Academy of Sciences. LX-2 and RAW 264.7 cells were cultured in DMEM medium (Gibco, China) supplemented with 10% fetal bovine serum (FBS) (Procell, China) and 1% penicillin/streptomycin (Solarbio, China). THP-1 cells were cultured in suspension in 1640 medium containing 10% FBS, 1% penicillin/streptomycin, and 0.05 mM β-mercaptoethanol. Prior to use, THP-1 cells were differentiated into adherent macrophages by treatment with 100 nM phorbol 12-myristate 13-acetate (PMA) for 24 h [27]. All cells were maintained and passaged in

a humidified incubator at 37°C with 5% $CO_2$. RAW 264.7 cells or PMA-induced differentiated THP-1 cells were treated with LPS, LPS + EgCF, or LPS + EgCF + GA for 12 h before WB analysis. LX-2 cells were treated with TGF-β1 (10 ng/mL) or EgCF (1 mg/mL) in combination with different concentrations of GA (10 nM, 20 nM) for 12 h before WB analysis. Cell-cell interactions were assessed using a conditioned medium crossover assay. LX-2 cells were treated for 12 h with the following conditioned media derived from THP-1 cells: CM1 (from LPS-stimulated THP-1 cells), CM2 (from THP-1 cells co-stimulated with LPS and EgCF), and CM2 co-incubated with GA. Normal culture medium was used as the control. THP-1 cells were treated for 12 h under the following conditions: conditioned medium from EgCF-stimulated LX-2 cells (CM), CM co-incubated with GA, and normal culture medium as the control. Cells were collected after treatment for WB analysis.

## SENP1 siRNA transfection

Three SENP1-targeting siRNAs (SENP1–1: 5′-CUGCCAUGUAUCUGCAUAUTT-3′; SENP1–2: 5′-GGCUCUGAUACUU CAUCAUTT-3′; SENP1–3: 5′-GACCAUUACACGCAAAGAUTT-3′) were synthesized by GenePharma. Transfection was performed using INTERFERin siRNA Transfection reagent (NoninBio, China). Briefly, 3 μL of siRNA was mixed with 4 μL of INTERFERin in 200 μL of serum-free medium, incubated for 10–15 min at room temperature to form complexes, and then added to THP-1 cells. After 6 h of transfection at 37°C, the medium was replaced with complete culture medium. Cells were further cultured for 24 h for RNA extraction and qRT-PCR validation. Based on the validation results, SENP1–1, which exhibited the highest knockdown efficiency, was selected for subsequent experiments (S1 Fig).

## Quantitative real-time PCR (qRT-PCR)

Total RNA was extracted from THP-1 cells using an RNA purification kit (Omega, USA) and reverse-transcribed into cDNA with the HiFiScript cDNA Synthesis Kit (Cwbio, China) according to the manufacturer's protocols. Quantitative real-time PCR (qRT-PCR) was performed on a CFX96 Touch detection system (Bio-Rad) using the UltraSYBR One-Step RT-qPCR Kit (Cwbio, China) with gene-specific primers (S3 Table). Genes expression levels were normalized to β-actin and calculated using the $2^{-\Delta\Delta Ct}$ method.

## Western blot analyses (WB)

After tissue or cell samples were lysed with RIPA lysate containing protease inhibitors, the supernatants were collected by centrifugation at 12,000 × g for 15 min. Protein concentration was determined using a BCA assay, and equal amounts of adjusted proteins were separated by 10% SDS-PAGE electrophoresis and transferred to PVDF membrane under constant pressure. After blocking with BSA for 2 h, membranes were incubated overnight at 4°C with the indicated primary antibodies (S4 Table), followed by incubation with secondary antibodies for 2 h at room temperature. Protein bands were visualized using enhanced chemiluminescence substrate, and band intensities were quantified with Image J software. Data are expressed as mean ± standard deviation.

## Cellular Immunofluorescence Assay

The THP-1 cells in different groups were successively fixed with 4% paraformaldehyde for 30 min, permeabilized with 0.3% Triton X-100 for 10 min, and blocked with 5% BSA for 30 min. Then, cells were incubated overnight at 4°C with primary antibodies (S5 Table), followed by a 2 h incubation with fluorescent secondary antibodies at room temperature in the dark. Nuclei were counterstained with DAPI, and images were acquired using a fluorescence microscope (NIKON ECLIPSE C1, Japan).

## Co-immunoprecipitation (Co-IP)

Following cell lysis on ice using RIPA lysis buffer containing a protease inhibitor cocktail, anti-Smad4 antibody and control antibodies were added to the lysates and incubated overnight to form protein-antibody complexes, while 10% of the lysate

was retained as a positive control; subsequently, protein A/G magnetic beads (RM02915, ABclonal, China) were added to the protein-antibody complexes and incubated at 4°C for 4 h to facilitate bead-antibody-protein complex binding; the beads were then centrifuged, the supernatant carefully aspirated, and the bead-protein complexes retained; the bead-protein complexes were then washed repeatedly to remove non-specifically bound material; finally, the washed beads were resuspended in 1 × SDS loading buffer and boiled at 100°C for 10 min; the resulting samples were then subjected to WB analysis according to standard protocols to detect the captured target proteins and assess the impact of each experimental factor on protein-protein interactions.

### Nuclear and cytoplasmic extraction

Nuclear and cytoplasmic proteins were isolated from THP-1 cells using a Nuclear and Cytoplasmic Protein Extraction Kit (Beyotime, China) according to the manufacturer's instructions, followed by WB to assess the expression levels of the target proteins in the nuclear and cytoplasmic fractions, where Histone H3 and GAPDH were used as loading controls for the nucleus and cytoplasm, respectively, and Smad4 served as the target protein.

### Statistical analysis

Data analysis and processing were performed using GraphPad Prism 8.0 software. The statistical significance between two groups was determined using an unpaired t-test, and comparisons among three or more groups were performed using one-way ANOVA. Data are reported as mean ± SD, with statistical significance set at $p < 0.05$: *$p < 0.05$, **$p < 0.01$, ***$p < 0.001$.

## Results

### Expression analysis of SUMOylation-associated proteins in hepatic lesions from CE patients

SUMOylation has been implicated in the progression of various fibrotic diseases [19,28]. However, its role in hepatic fibrosis caused by CE remains unclear. To investigate this, paraffin-embedded liver samples from CE patients were collected and categorized into two groups based on their proximity to the hydatid cyst: peri-lesion (PL) tissue (adjacent to the cyst wall) and adjacent normal (AN) tissue (approximately 5 cm from the lesion) [29]. Histopathological staining revealed more pronounced inflammatory cell infiltration and collagen fiber deposition in PL tissues compared to AN tissues, confirming the presence of fibrotic lesions. Immunohistochemical analysis was further performed to examine the expression of key SUMO pathway components. The results showed that the expression levels of SUMO1, Ubc9, and the fibrosis markers α-SMA and COL1A1 were significantly higher in fibrotic PL regions than in AN regions. Conversely, the expression of the deSUMOylase SENP1 was markedly downregulated in PL tissues (Fig 2). Collectively, these data indicate a significant dysregulation of the SUMOylation in the fibrotic niche of CE patients, suggesting that SUMOylation may be involved in the pathogenesis of CE-associated hepatic fibrosis.

### GA attenuates hepatic fibrosis and modulates the expression profile of SUMOylation related proteins in a mice model of *Echinococcus granulosus* infection

Based on the association between the dysregulation of the SUMOylation pathway and fibrosis observed in clinical samples, we speculate that targeting this pathway may provide therapeutic ideas for CE liver fibrosis. GA has been reported to have SUMOylation inhibitory activity [30–31], we evaluated its anti-fibrotic efficacy and regulatory effect on the SUMOylation in a CE mouse model. The model was established by subcapsular hepatic injection of PSCs. Daily intraperitoneal GA (25 mg/kg) was administered for 4 weeks starting from week 8 post-infection (Fig 3a). In addition, uninfected mice treated with the same dose of GA were used as the drug toxicity control group. Compared with the CE group, GA treatment significantly reduced hepatic cysts volume (by approximately 42.36%; $P < 0.05$; Fig 3b and 3c) and decreased serum levels of the key pro-fibrotic cytokine TGF-β1 ($P < 0.01$; Fig 3d).

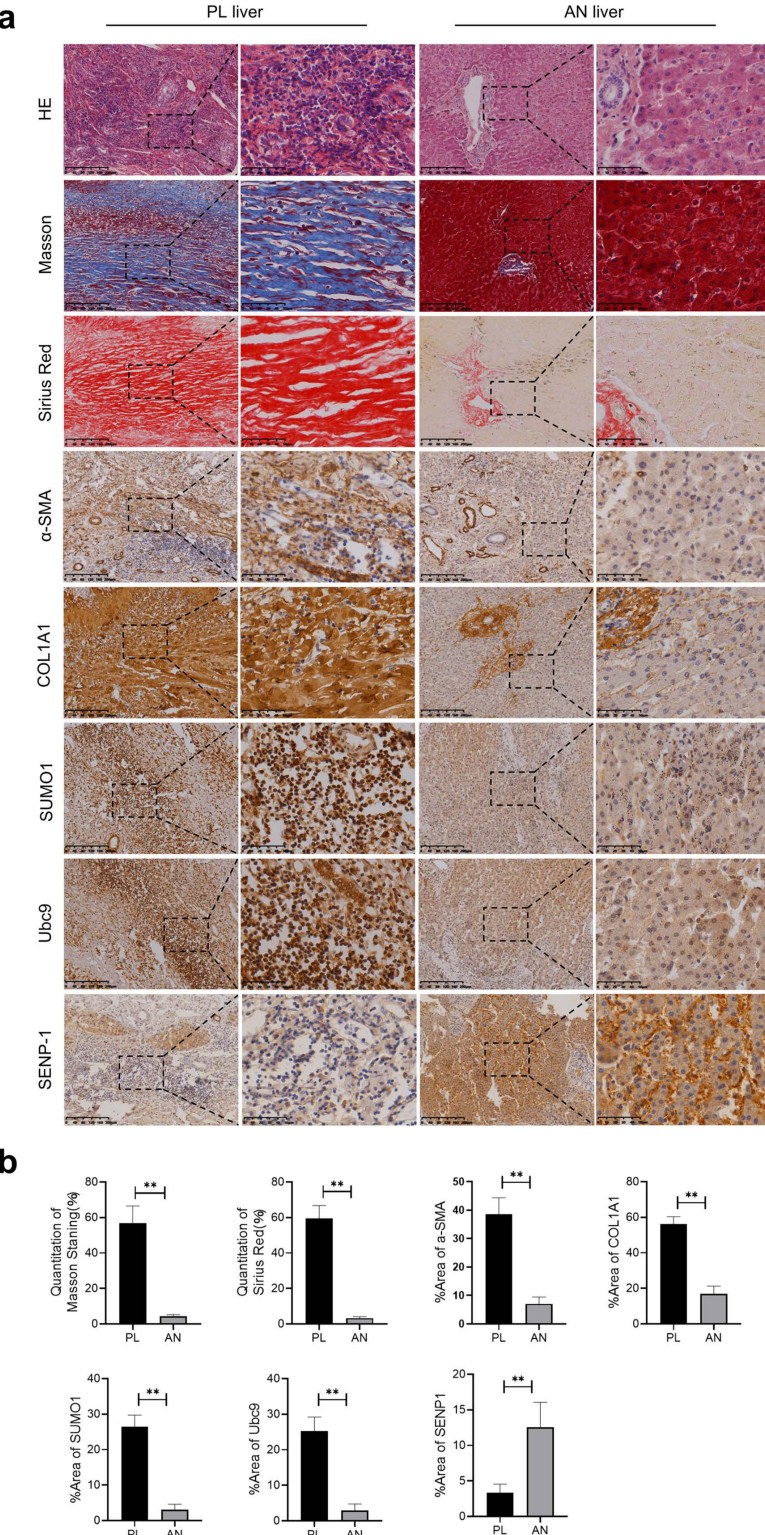

**Fig 2. Differential expression of SUMO proteins in the lesion microenvironment of hepatic CE patients. (a)** Representative H&E, Masson, Sirius red and immunohistochemical (IHC) staining of a-SMA, COL1A1, SUMO1, Ubc9 and SENP1 in peri-lesion (PL) liver and adjacent normal (AN) liver obtained from hepatic CE patients. **(b)** Quantitative analysis of positively stained areas for Masson staining, Sirius red and α-SMA, COL1A1, SUMO1, Ubc9, SENP1 expression. Data represent mean ± SD. n = 5, statistical significance: *p < 0.05, **p < 0.01 and ***p < 0.001.

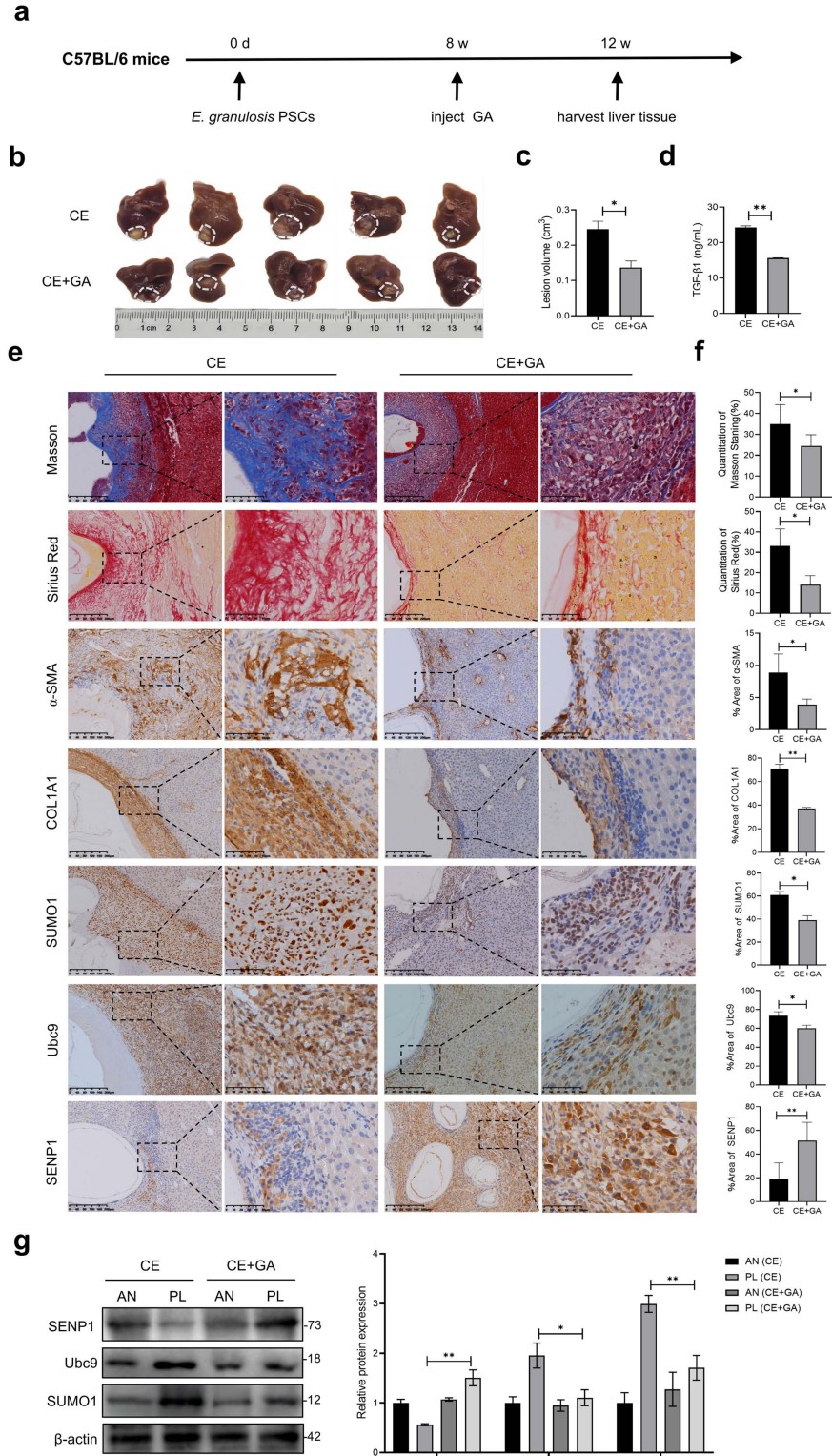

**Fig 3. GA modulated key SUMO proteins *in vivo*. (a)** Mice injected with PSCs under the liver capsule were administered GA intraperitoneally at 8 weeks, and the experiment concluded at 12 weeks. **(b)** Gross images of the livers of mice in each group. **(c)** Lesion volume were calculated and compared among two groups. **(d)** Serum TGF-β1 levels of the two groups were measured by ELISA. **(e)** Representative Masson, Sirius red and IHC staining

of a-SMA, COL1A1, SUMO1, Ubc9 and SENP1 in peri-lesion (PL) liver from two groups. **(f)** Quantitative analysis of positively stained areas for Masson staining, Sirius red and α-SMA, COL1A1, SUMO1, Ubc9, SENP1 expression. **(g)** Western blot (WB) analysis of the protein expression levels of SUMO1, Ubc9, SENP1 in peri-lesion (PL) or adjacent normal (AN) liver from two groups mice. Data represent mean ± SD. n = 5, statistical significance: *p < 0.05, **p < 0.01 and ***p < 0.001.

Histopathological analysis revealed extensive collagen deposition in the livers of CE mice, with increased positive areas in Masson's trichrome (34.87%) and Sirius Red (33.11%) staining, along with elevated expression of the fibrosis markers α-SMA and COL1A1. These fibrotic alterations were markedly ameliorated by GA intervention (Fig 3e and 3f). Notably, consistent with findings in human tissues, hepatic tissues from the CE model group exhibited up-regulation of SUMO1 and Ubc9, and down-regulation of SENP1. GA treatment reversed this expression profile (Fig 3e and 3g). Moreover, histologic examination of major organs from the GA-only control group showed no significant differences from the normal control group (S2 Fig), indicating no observable organ toxicity under the experimental conditions.

In summary, GA treatment not only effectively alleviated the pathological features of hepatic fibrosis but also reversed the changes in the SUMOylation within fibrotic lesions in the CE mouse model.

## GA modulates the inflammatory responses of macrophages induced by EgCF

The aforementioned *in vivo* experiments indicated that the anti-fibrotic effect of GA is associated with the modulation of the SUMOylation. Based on the central role of macrophages in the progression of fibrosis and the positive correlation between M2 polarization and fibrosis [32,33], we hypothesized that GA might exert its function by regulating macrophage activity. To test this hypothesis, we first performed validation in the animal model. Immunofluorescence analysis revealed that GA treatment markedly reduced the infiltration of SUMO1-positive/ F4/80-positive macrophages in the infected livers (Fig 4a). Furthermore, we established an *in vitro* inflammation model using THP-1 and RAW264.7 macrophage cell lines stimulated with EgCF. WB analysis showed that EgCF stimulation upregulated M2 polarization marker CD206 while downregulating M1 marker CD86 expression in THP-1 cells. GA intervention markedly inhibited the EgCF-induced upregulation of CD206 (Fig 4b). A consistent trend was observed in RAW264.7 cells (Fig 4c). These results suggest that GA can inhibit the macrophages polarization toward a pro-fibrotic phenotype in both *in vivo* and *in vitro* models of CE.

## GA inhibits EgCF induced Smad4 SUMOylation

The above studies found that GA regulates macrophage polarization. Since the TGF-β/Smad signaling pathway is one of the key ways to regulate macrophage polarization [34], and SUMOylation can regulate the activity and nuclear translocation of Smad4 [21], we further explored whether GA exerts its effect by affecting the SUMOylation of Smad4. Firstly, total protein was extracted from the liver tissues of mice in each group. The WB results showed that, compared with the CE group, the protein expressions of SUMO1 and Smad4 decreased after GA treatment; the CO-IP results further indicated that GA inhibited the CE-induced Smad4 SUMOylation modification (Fig 5a). *In vitro*, we established an EgCF-induced macrophage inflammation model. Western blot analysis demonstrated that compared with the EgCF + LPS-treated group, GA intervention significantly downregulated the protein expression of SUMO1, Ubc9, and Smad4, while upregulating the deSUMOylating enzyme SENP1 (Fig 5b and 5c). CO-IP experiments further confirmed that GA significantly suppressed the EgCF-induced interaction between Smad4 and SUMO1 (Fig 5d and 5e). Furthermore, cellular immunofluorescence and nuclear-cytoplasmic fractionation experiments indicated that GA treatment reduced the nuclear localization of Smad4 (Fig 5f and 5g). To clarify the role of SENP1 in this process, we knocked down SENP1 expression in THP-1 cells. The results showed that SENP1 knockdown attenuated the EgCF-induced Smad4 SUMOylation, while the ability of GA to reverse this effect was significantly enhanced, indicating that the upregulation of SENP1 is a critical mechanism by which GA inhibits Smad4 SUMOylation (Fig 5h). Taken together, these results suggest that GA may inhibit the Smad4

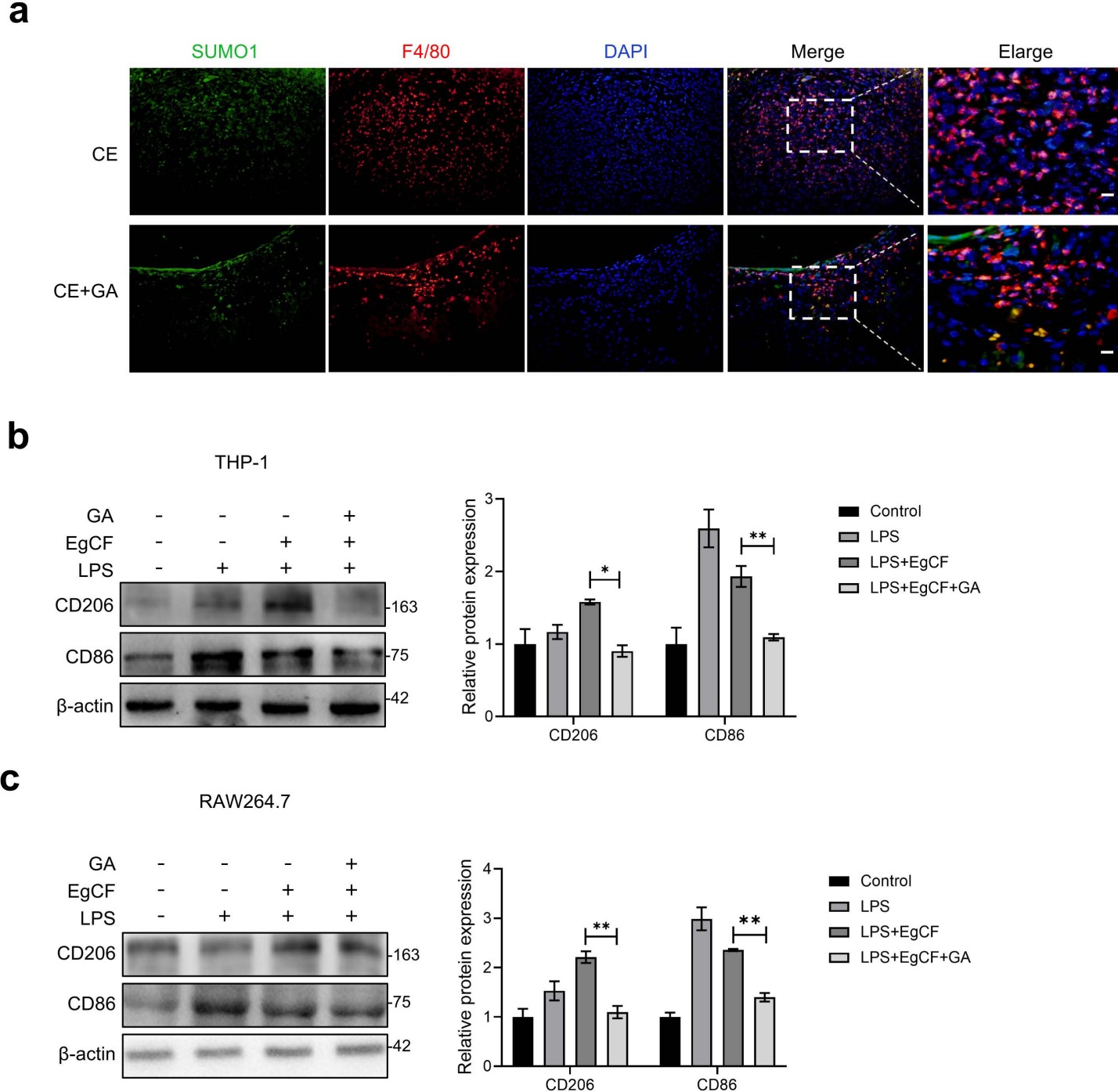

**Fig 4. GA inhibited macrophage polarization induced by EgCF. (a)** Immunofluorescence (IF) staining for SUMO1 (green) and F4/80 (red) were performed on Paraffin liver tissue sections, and the nuclei were stained with DAPI (blue), scale bar: 20 μm.RAW 264.7 cells and THP-1 cells were pretreated with LPS (1 μg/mL) and then with EgCF (1 mg/mL) and GA (10 nM) for 12 h, CD206 and CD86 protein expression levels in RAW 264.7 cells **(b)** and THP-1 cells **(c)** were detected by WB. Data are presented as mean±SEM of three independent experiments. Statistical significance: *p < 0.05, **p < 0.01 and ***p < 0.001.

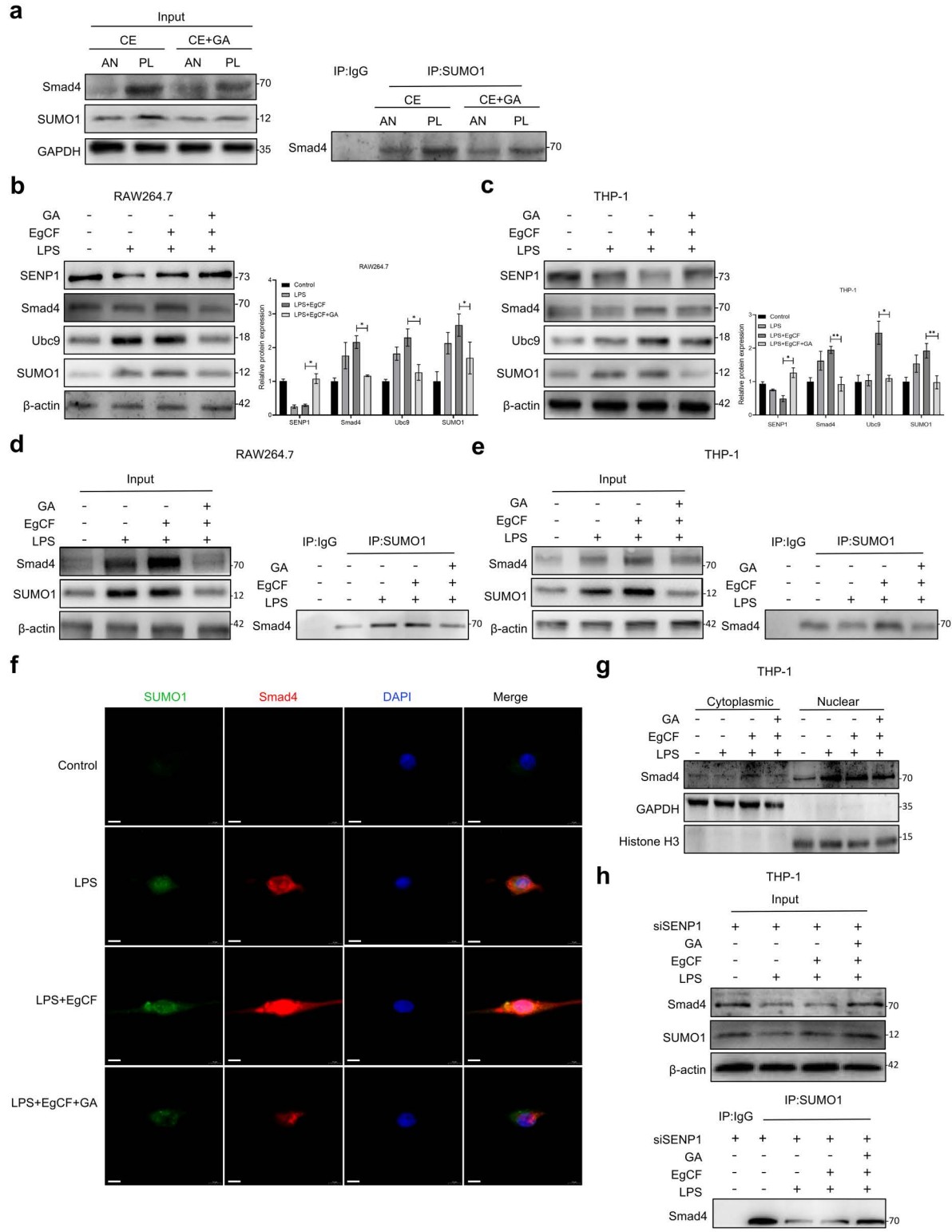

**Fig 5. GA inhibited EgCF induced Smad4 SUMOylation. (a)** SUMO1-Smad4 association in peri-lesion (PL) and adjacent normal (AN) liver from two groups mice: CO-IP with anti-SUMO1 antibody or Mouse IgG control, followed by Smad4-specific detection (WB), with Input controls showing both proteins. RAW 264.7 cells and THP-1 cells were pretreated with LPS (1 μg/mL) and then with EgCF (1 mg/mL) and GA (10 nM) for 12 h, SENP1, Smad4,

Ubc9 and SUMO1 protein expression levels in RAW 264.7 cells **(b)** and THP-1 cells **(c)** were detected by WB. SUMO1-Smad4 association in RAW 264.7 cells **(d)** and THP-1 cells **(e)** different groups: CO-IP with anti-SUMO1 antibody or Mouse IgG control, followed by Smad4-specific detection (WB), with Input controls showing both proteins. **(f)** Immunofluorescence double staining of SUMO1 (green) and Smad4 (red) in differentially treated THP-1 cell groups. Nuclei were counterstained with DAPI (blue), scale bar: 10 μm. **(g)** Western blot for Smad4 expression in cytoplasmic and nuclear fractions of THP-1 cells treated with LPS (1 μg/mL) and then with EgCF (1 mg/mL) and GA (10 nM) for 12 **h. (h)** The impact of SENP1 knockdown on Smad4 SUMOylation in THP-1 cells was analyzed by CO-IP. Data are presented as mean±SEM of three independent experiments. Statistical significance: *p<0.05, **p<0.01 and ***p<0.001.

SUMOylation by upregulating SENP1, thereby reducing its nuclear translocation, and thereby inhibiting macrophage polarization towards the pro-fibrotic M2 phenotype.

## GA inhibited the EgCF-induced HSCs activation and its crosstalk with macrophage polarization

To further investigate the involvement of GA in EgCF-induced hepatic stellate cell (HSC) activation and proliferation, we treated TGF-β1- or EgCF-stimulated LX-2 cells with varying GA concentrations (0, 10 nM, 20 nM) for 12 hours. Western blot analysis revealed that GA significantly suppressed the expression of proliferation marker PCNA and activation marker α-smooth muscle actin (α-SMA) in LX-2 cells (Fig 6a and 6b). Subsequently, LX-2 cells were co-cultured with THP-1 macrophage-derived conditioned medium (CM), where both CM1 (supernatant from LPS-stimulated THP-1 cells) and CM2 (supernatant from LPS+EgCF-stimulated THP-1 cells) were found to markedly upregulate PCNA and α-SMA levels in LX-2 cells. Notably, GA treatment specifically counteracted CM2-induced pro-fibrotic effects (Fig 6c). When THP-1 macrophages were co-cultured with LX-2-derived CM (supernatant from EgCF-stimulated LX-2 cells), GA significantly downregulated the expression of M2 polarization marker CD206 and macrophage activation marker CD86 in THP-1 cells (Fig 6d). Furthermore, GA effectively suppressed the secretion of IL-13 by LX-2 cells and TGF-β1 by THP-1 cells (Fig 6e). These findings collectively suggest that GA may exert its anti-fibrotic effects through bidirectional regulation of macrophage-HSC crosstalk, as evidenced by its capacity to attenuate both HSC activation/proliferation and macrophage polarization/activation in this reciprocal cellular communication system.

## 4. Discussion

Our research results show that GA ameliorates *E.granulosus* infection-mediated hepatic fibrosis by specifically inhibiting SUMOylation of Smad4 in macrophages through targeted modulation of the SUMOylation pathway, thereby hindering pathogenic macrophage-hepatic stellate cell crosstalk. This discovery not only reveals the central pathogenic driver role of the SUMO-Smad4 axis in parasite-induced fibrogenesis, but also provides a novel therapeutic target and candidate drug for anti-fibrotic treatment in hepatic echinococcosis.

Chronic Th2 inflammation triggered by *E.granulosus* persistently activates HSCs, causing them to transform into myofibroblasts, over-synthesize extracellular matrix (ECM), and abnormally deposit around the lesion, forming fibrous scars [35–36]. Recent studies demonstrate that SUMOylation and deSUMOylation play important roles in the process of heart, lung, liver and kidney fibrosis [19,22,28,37–39]. Our immunohistochemical results showed that in the lesion microenvironment of patients with cystic hepatic echinococcosis and infected mice, the expression of key molecules SUMO1, Ubc9 and SENP1 in the SUMOylation pathway was abnormal, suggesting that the imbalance of SUMOylation modification is a characteristic pathological event in the fibrotic microenvironment of hepatic echinococcosis. GA, as a natural SUMOylation inhibitor, exhibits therapeutic potential for counteracting organ fibrosis [22,28]. Interestingly, GA significantly modulates expression levels of SUMO1, Ubc9, and SENP1 in *E. granulosus* infection-driven hepatic fibrosis models, further indicating the SUMOylation pathway as a key hub of the anti-fibrotic effect of GA.

Our further research found that in the mouse model of *E.granulosus* infection, SUMO1 had significant spatial co-localization with the macrophage-specific marker F4/80. *In vitro* experiments further confirmed that EgCF stimulation

PLOS Neglected Tropical Diseases

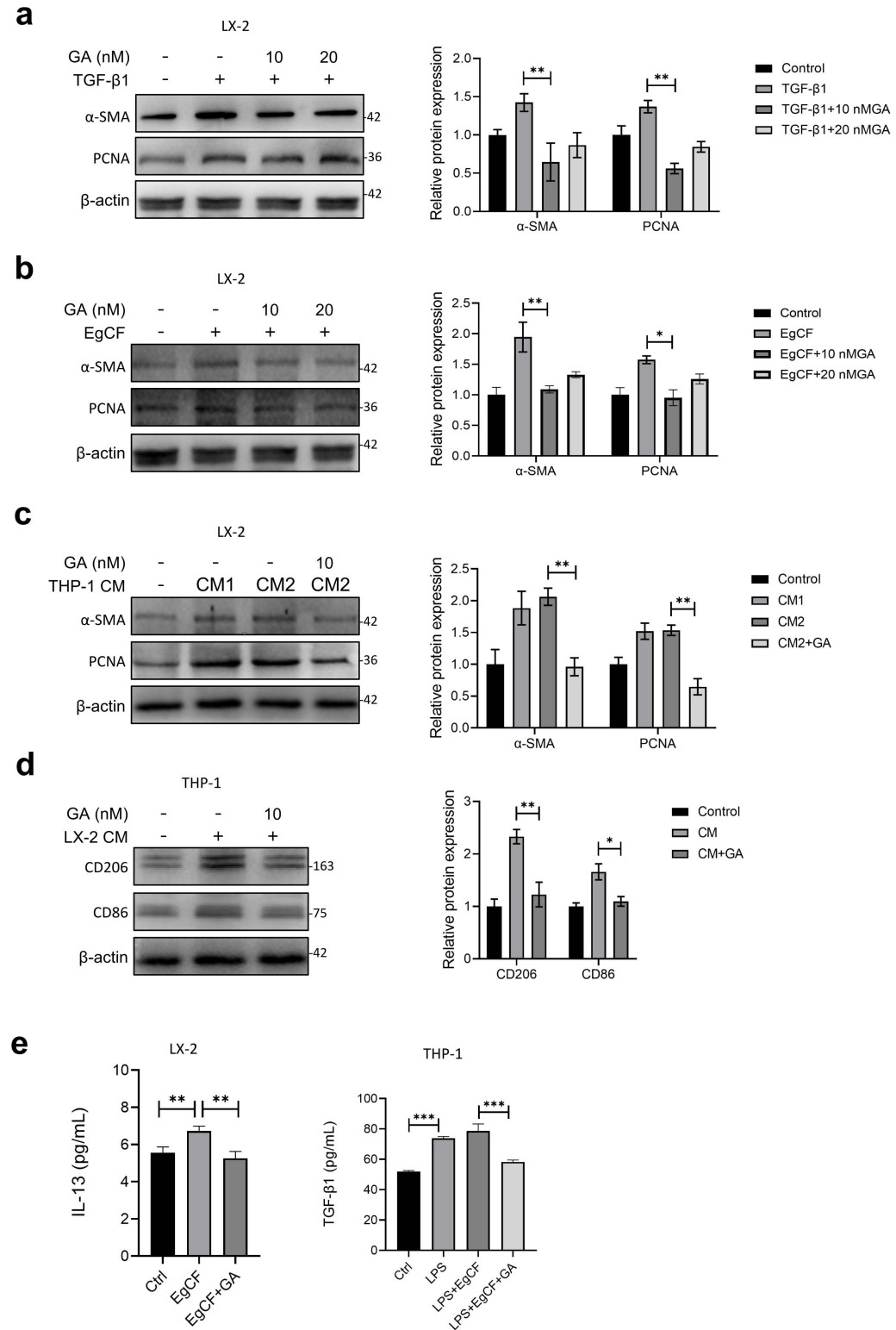

**Fig 6. GA inhibited the EgCF-induced HSCs activation and its crosstalk with macrophage polarization. (a, b)** LX-2 cells were treated with TGF-β1 (10 ng/mL) or EgCF (1 mg/mL) and GA (10 nM, 20 nM) for 12 h, the α-SMA and PCNA protein levels were analyzed by WB. **(c)** LX-2 cells were treated with conditioned medium (CM) from LPS-stimulated THP-1 cell ssupernatant (CM1) or LPS + EgCF-stimulated THP-1 cell ssupernatant (CM2) or

CM2 and GA (10 nM) for 12 h, the α-SMA and PCNA protein levels were analyzed by WB. **(d)** THP-1 cells were treated with conditioned medium (CM) from EgCF-stimulated LX-2 cells ssupernatant for 12 h, the CD206 and CD86 protein levels were analyzed by WB. **(e)** Cell supernatant IL-13, TGF-β1 levels were measured by ELISA. Data are presented as mean±SEM of three independent experiments. Statistical significance: *p<0.05, **p<0.01 and ***p<0.001.

could simultaneously up-regulate the expressions of SUMO1, E2 binding enzyme UBC9 and M2-type polarization marker CD206 in macrophages. However, GA intervention not only reduces the protein level of SUMO1, but also significantly weakens the co-localization signal of SUMO1 and F4/80, effectively reversing the M2-type hyperpolarization phenotype of macrophages induced by EgCF. Previous studies have shown that SUMOylation modification drives the core mechanism of the M2 polarization process in macrophages by dynamically regulating key signaling proteins, thereby mediating patho-physiological responses such as tissue repair, neuroprotection, and immunosuppression [40–43]. Therefore, we propose that GA inhibits the fibrosis process by targeting the SUMOylation modification pathway of macrophages and regulating their immunophenotype.

The molecular mechanism by which SUMOylation drives the fibrotic process by targeting and regulating the core components of the TGF-β/Smad pathway (SUMOylation of TGFβRI enhances the phosphorylation of Smad2/3, SUMOylation of Smad2 increases its phosphorylation level and transcriptional activity, and SUMOylation of Smad4 mediates ECM remodeling) has been clarified [44–46]. GA is a negative regulator of the SUMOylation modification of Smad4. In our experiment, *E.granulosus* infection significantly enhanced the SUMOylation modification of Smad4, increased the level of serum TGF-β1, and exacerbated fibrosis. GA intervention not only significantly inhibits the SUMOylation modification of Smad4, but also hinders the nuclear translocation process of SUMOylation Smad4 in macrophages induced by EgCF (verified by the nuclear-cytoplasmic separation experiment), thereby reversing the M2-type polarization phenotype (CD206↓). And synergistically inhibit the proliferation and activation of HSCs (α-SMA/ PCNA↓). To sum up, the SUMO modification of Smad4 may be the key factor for GA to inhibit macrophage polarization and resist echinococcosis fibrosis.

This study reveals that SUMOylation of Smad4 regulates the M2 polarization of macrophages and the crosstalk between macrophages and HSCs, thereby promoting the progression of hepatic fibrosis mediated by *E.granulosus* infection. This finding not only provides an important theoretical basis for a deeper understanding of the pathogenesis of parasitic liver fibrosis, but also proposes Smad4 SUMOylation as a new target for intervention with great potential for anti-fibrosis, and finds that GA as an effective inhibitor of the target can significantly inhibit the above pathogenic pathways and exert anti-fibrosis effects. It provides innovative strategies and drug candidates for anti-fibrotic treatment of hydatid disease.

The study still has the following limitations. Firstly, in terms of pharmacokinetics, the *in vivo* course of GA in infection models has not been defined. Although the existing dosing regimens refer to previous studies and simulate the clinical treatment window, their local distribution, metabolism and effective duration of action still need to be systematically evaluated, which is crucial to optimize the dosing strategy and promote the clinical translation. Second, in terms of mechanism specificity, the direct inhibitory effect of GA on Smad4 SUMOylation still needs to be further verified. Although SENP1 knockdown experiments suggest that this enzyme is involved in regulation, as a multi-targeted compound, whether GA indirectly affects SUMOylation through other pathways has not been completely ruled out. At the same time, the key signaling molecules mediating the pathogenic dialogue between macrophages and HSCs are still not clear, which restricts the development of more targeted intervention strategies. Although animal experiments showed that GA could reverse the dysregulation of SUMOylation at the tissue level and reduce the infiltration of SUMO1-positive macrophages, indirectly supporting the proposed regulatory axis *in vitro*, there is still a lack of evidence to directly validate the key steps of the pathway in specific cell types *in vivo*. In the future, through the construction of site-specific mutation models, multi-omics screening and cell-specific gene manipulation animal models, the cell-specific role and pathological significance of this pathway can be further confirmed *in vivo*, and promote its transformation into targeted therapy.

## Conclusions

Taken together, the present study clarified that the imbalance of SUMOylation is a central molecular hub in the progression of liver fibrosis induced by *E. granulosus* infection. GA can specifically inhibit the SUMOylation of Smad4 and its nuclear translocation by targeting this pathway, thereby effectively reversing the M2 macrophage over-polarization and inhibiting the activation and proliferation of hepatic stellate cells. More importantly, GA significantly disrupted the vicious cycle of mutual promotion between M2-type macrophages and activated HSCs. Therefore, GA exerts its anti-hydatid liver fibrosis effect through multi-target intervention of SUMOylation - Smad4 signaling axis and cell-cell interaction network, which provides an important experimental basis for targeted therapy based on SUMOylation regulation.

## Supporting information

**S1 Fig. Interference efficiency of SENP-1 was qualified by qRT-PCR.** THP-1 cells were transfected with control siRNA (ctrl) or three SENP1-targeting siRNAs (siSENP1–1, siSENP1–2, and siSENP1–3), respectively. (a) At 24 hours post-transfection, the mRNA expression level of SENP1 was detected by qRT-PCR. Data are presented as the mean ± SEM of three independent experiments. Statistical significance: $*p < 0.05$, $**p < 0.01$, $***p < 0.001$.
(TIFF)

**S2 Fig. Safety evaluation of GA in mice.** (a) HE staining results of paraffin sections of the main organs (heart, liver, spleen, lung, and kidney) in the GA group and the normal control group.
(TIFF)

**S1 Table. Antibodies information used in immunohistochemicaly analysis.**
(DOCX)

**S2 Table. Antibodies information used intissue immunofluorescence analysis.**
(DOCX)

**S3 Table. Primer sequences for qRT-PCR.**
(DOCX)

**S4 Table. Antibodies information used in Western blot analysis.**
(DOCX)

**S5 Table. Antibodies information used in cellular immunofluorescence assay.**
(DOCX)

## Author contributions

**Conceptualization:** Xiangwei Wu, xueling chen.

**Data curation:** Qiuyue Chen, Xueting Yu.

**Formal analysis:** Qiuyue Chen, Xinyu Jiang.

**Funding acquisition:** Dan Dong, Huijiao Jiang, Xiangwei Wu, xueling chen.

**Investigation:** Qiuyue Chen, Dan Dong, Xueting Yu, Xinyu Jiang, Huijiao Jiang, Jun Hou, Lianghai Wang, Junying Xu, Xiangwei Wu, xueling chen.

**Methodology:** Qiuyue Chen, Dan Dong, Jun Hou, Lianghai Wang, Xiangwei Wu, xueling chen.

**Project administration:** Xiangwei Wu, xueling chen.

**Resources:** Xiangwei Wu, xueling chen.

**Software:** Lianghai Wang, Junying Xu.

**Supervision:** Jun Hou, Lianghai Wang, Xiangwei Wu, xueling chen.

**Validation:** Qiuyue Chen, Dan Dong, xueling chen.

**Visualization:** Qiuyue Chen, Dan Dong.

**Writing – original draft:** Qiuyue Chen, Dan Dong.

**Writing – review & editing:** Qiuyue Chen, Dan Dong, Xiangwei Wu, xueling chen.

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
