## [Decision Letter · Decision Letter 0]

7 Oct 2025

Ginkgolic acid attenuates e chinococcus granulosus  infection-induced hepatic fibrosis by inhibiting Smad4 SUMOylation

Dear Dr. chen,

Thank you for submitting your manuscript to PLOS Neglected Tropical Diseases. After careful consideration, we feel that it has merit but does not fully meet PLOS Neglected Tropical Diseases's publication criteria as it currently stands. Therefore, we invite you to submit a revised version of the manuscript that addresses the points raised during the review process.

Please submit your revised manuscript within 60 days Dec 06 2025 11:59PM. If you will need more time than this to complete your revisions, please reply to this message or contact the journal office at plosntds@plos.org. Please include the following items when submitting your revised manuscript:

We look forward to receiving your revised manuscript.

Kind regards,

Robert Adamu SHEY, Ph.D.

Guest Editor

Jong-Yil Chai

Section Editor

Shaden Kamhawi

co-Editor-in-Chief

Paul Brindley

co-Editor-in-Chief

**Additional Editor Comments:**

Dear Authors,

Kindly revise the manuscript, taking consideration all comments from the reviewers.

Kind regards

**Journal Requirements:**

At this stage, the following Authors/Authors require contributions: Qiuyue Chen, Dan Dong, Xueting Yu, Xinyu Jiang, Huijiao Jiang, Jun Hou, Lianghai Wang, Junying Xu, Xiangwei Wu, and xueling chen. Please ensure that the full contributions of each author are acknowledged in the "Add/Edit/Remove Authors" section of our submission form.

2) We have noticed that you have uploaded Supporting Information files, but you have not included a list of legends. Please add a full list of legends for your Supporting Information files after the references list.

3) Some material included in your submission may be copyrighted. According to PLOSu2019s copyright policy, authors who use figures or other material (e.g., graphics, clipart, maps) from another author or copyright holder must demonstrate or obtain permission to publish this material under the Creative Commons Attribution 4.0 International (CC BY 4.0) License used by PLOS journals. Please closely review the details of PLOSu2019s copyright requirements here: PLOS Licenses and Copyright. If you need to request permissions from a copyright holder, you may use PLOS's Copyright Content Permission form.

Potential Copyright Issues:

i) Please confirm (a) that you are the photographer of 1B, or (b) provide written permission from the photographer to publish the photo(s) under our CC BY 4.0 license.

ii) Figure 1A. Please confirm whether you drew the images / clip-art within the figure panels by hand. If you did not draw the images, please provide (a) a link to the source of the images or icons and their license / terms of use; or (b) written permission from the copyright holder to publish the images or icons under our CC BY 4.0 license. Alternatively, you may replace the images with open source alternatives. See these open source resources you may use to replace images / clip-art:

4) In the online submission form, you indicated that Data will be made available on request.. All PLOS journals now require all data underlying the findings described in their manuscript to be freely available to other researchers, either

1. In a public repository

2. Within the manuscript itself

3. Uploaded as supplementary information.

2) If any authors received a salary from any of your funders, please state which authors and which funders..

**Reviewers' Comments:**

Reviewer's Responses to Questions

**Key Review Criteria Required for Acceptance?**

**Methods**

-Are the objectives of the study clearly articulated with a clear testable hypothesis stated?

-Is the study design appropriate to address the stated objectives?

-Is the population clearly described and appropriate for the hypothesis being tested?

-Is the sample size sufficient to ensure adequate power to address the hypothesis being tested?

-Were correct statistical analysis used to support conclusions?

-Are there concerns about ethical or regulatory requirements being met?

Reviewer #1: (No Response)

Reviewer #2: (No Response)

**Results**

-Does the analysis presented match the analysis plan?

-Are the results clearly and completely presented?

-Are the figures (Tables, Images) of sufficient quality for clarity?

Reviewer #1: (No Response)

Reviewer #2: (No Response)

**Conclusions**

-Are the conclusions supported by the data presented?

-Are the limitations of analysis clearly described?

-Do the authors discuss how these data can be helpful to advance our understanding of the topic under study?

-Is public health relevance addressed?

Reviewer #1: (No Response)

Reviewer #2: (No Response)

**Editorial and Data Presentation Modifications?**

Reviewer #1: (No Response)

Reviewer #2: Please enhance the clarity of the bar charts in Figures 1 and 2�Please improve the resolution of all result figures in the manuscript�.

Please display the sample size for each group in the article's figures.

The SUMO1 Western blot band in Figure 4a appears blurred.

Some figures like e.g., immunofluorescence images lack scale bars .

Some references lack DOI or PMID numbers (e.g., Ref 3).

The inconsistent use of "Ginkgo acid"(Line 89) and "Ginkgolic acid"(Title) should be unified to "Ginkgolic acid (GA)".

**Summary and General Comments**

Reviewer #1: This manuscript investigates the therapeutic potential of GA in alleviating liver fibrosis induced by Echinococcus granulosus infection, focusing on its inhibition of Smad4 SUMOylation and modulation of macrophage–hepatic stellate cell crosstalk. The study is well-structured, combining clinical samples, animal models, and in vitro experiments to comprehensively explore the molecular mechanism.

While the findings are promising and suggest GA as a potential anti-fibrotic agent, the manuscript requires revisions to enhance clarity, statistical rigor, and methodological detail.

For the Introduction Section

While the introduction touches upon multiple concepts—CE, GA, SUMOylation, macrophage polarization, and TGF-β/Smad signaling—the connections between these ideas are not seamlessly established. Although the research gap is mentioned, it should be more explicitly stated what specific unanswered question the study aims to address.

For the Methods Section

The authors please provide missing critical details regarding experimental procedures, statistical reporting, and cell culture conditions.

For the Results Section

The language often implies a direct and proven causal relationship where the data primarily show correlation. The narrative flow between some paragraphs is abrupt.

the Results section must be substantially revised to include essential statistical details, employ more precise and cautious language regarding mechanistic claims, and improve overall clarity and language quality.

For the Discussion Sections

While a limitations paragraph is included, it remains superficial. The authors should delve deeper: Pharmacokinetics/Pharmacodynamics Specificity of GA In vivo validation.

Reviewer #2: In the animal experiment, the purpose of the "GA safety group" is unclear; it is recommended to clarify its role as a drug toxicity control group.Additionally, detailed justification for initiating GA administration at the 8th week is lacking.

The mechanism by which GA inhibits Smad4 SUMOylation in Figure 4 lacks direct evidence (e.g., SENP1 overexpression/knockdown experiments).

Control experiments demonstrating GA's specificity in inhibiting SUMOylation are absent (e.g., comparison with other SUMO inhibitors).

PLOS authors have the option to publish the peer review history of their article (what does this mean? ). If published, this will include your full peer review and any attached files.

**Do you want your identity to be public for this peer review?** For information about this choice, including consent withdrawal, please see our Privacy Policy .

Reviewer #1: No

Reviewer #2: No

**Figure resubmission:**
---

## [Editor Report · Decision Letter 1]

6 Jan 2026

Dear Dr. chen,

We are pleased to inform you that your manuscript 'Ginkgolic acid attenuates e chinococcus granulosus  infection-induced hepatic fibrosis by inhibiting Smad4 SUMOylation' has been provisionally accepted for publication in PLOS Neglected Tropical Diseases.

Best regards,

Robert Adamu SHEY, Ph.D.

Guest Editor

Jong-Yil Chai

Section Editor

Shaden Kamhawi

co-Editor-in-Chief

Paul Brindley

co-Editor-in-Chief

---

## [Editor Report · Acceptance letter]

Dear Dr. chen,

We are delighted to inform you that your manuscript, "Ginkgolic acid attenuates echinococcus granulosus infection-induced hepatic fibrosis by inhibiting Smad4 SUMOylation," has been formally accepted for publication in PLOS Neglected Tropical Diseases.

Best regards,

Shaden Kamhawi

co-Editor-in-Chief

Paul Brindley

co-Editor-in-Chief
